# Effect of Scanned Area and Operator on the Accuracy of Dentate Arch Scans with a Single Implant

**DOI:** 10.3390/jcm11144125

**Published:** 2022-07-15

**Authors:** Vinicius Rizzo Marques, Gülce Çakmak, Hakan Yilmaz, Samir Abou-Ayash, Mustafa Borga Donmez, Burak Yilmaz

**Affiliations:** 1Department of Reconstructive Dentistry and Gerodontology, School of Dental Medicine, University of Bern, 3012 Bern, Switzerland; vinicius.rizzo-marques@zmk.unibe.ch (V.R.M.); guelce.cakmak@unibe.ch (G.Ç.); samir.abou-ayash@unibe.ch (S.A.-A.); burak.yilmaz@unibe.ch (B.Y.); 2İkon Oral and Dental Health Center, Istanbul 34275, Turkey; hakanyilmaz90@gmail.com; 3Department of Prosthodontics, Faculty of Dentistry, Istinye University, Istanbul 34010, Turkey; 4Department of Restorative, Preventive, and Pediatric Dentistry, School of Dental Medicine, University of Bern, 3012 Bern, Switzerland; 5Division of Restorative and Prosthetic Dentistry, The Ohio State University, Columbus, OH 43210, USA

**Keywords:** implant scan, operator, precision, scan area, trueness

## Abstract

Studies have shown the effect of the operator and scanned areas on the accuracy of single implant scans. However, the knowledge on the scan accuracy of the remaining dental arch during single implant scans, which may affect the occlusion, is limited. The aim of this study was to investigate the effect of scanned areas and the operator on the scan accuracy of a dentate arch while scanning a single implant. A dentate model with an anterior implant was digitized with a laboratory scanner (reference scan). Three operators with similar experience performed 10 complete- and 10 partial-arch scans (left 2nd molar to right canine) with an intraoral scanner (TRIOS 3), and these scans were superimposed over the reference. The accuracy was analyzed at 22 points in complete-arch and at 16 points in partial-arch scans on 2nd molars and incisors. Data were evaluated with 2-way ANOVA and Tukey HSD tests (α = 0.05). The trueness of the total scanned area was higher in partial- than in complete-arch scans (*p* < 0.001). The trueness and precision of the scans were higher in the anterior site compared with the posterior in complete- (trueness: *p* ≤ 0.022, precision: *p* ≤ 0.003) and partial-arch (trueness: *p* ≤ 0.016, precision: *p* ≤ 0.016) scans of each operator and when the operator scan data were pooled. The complete-arch scan’s precision was not influenced by the operator (*p* ≥ 0.029), whereas the partial-arch scans of operator 1 and 2 were significantly different (*p* = 0.036). Trueness was higher in partial- compared with complete-arch scans, but their precision was similar. Accuracy was higher in the anterior site regardless of the scan being a partial- or a complete-arch. The operator’s effect on the accuracy of partial- and complete-arch scans was small.

## 1. Introduction

The launch of new intraoral scanners (IOSs) and improved accuracy with scanner technologies enabled the fabrication of implant-supported monolithic crowns through a direct digital workflow with clinically acceptable accuracy and less patient discomfort in recent years [1,2,3,4,5,6,7]. However, studies focusing on varying clinical scenarios are still conducted to identify the factors affecting the scan accuracy [7,8,9,10,11] such as scanned area and operator experience [11,12,13,14,15,16,17,18,19]. Regarding the effect of operator experience on scan accuracy, conflicting results have been reported [18,20,21]. Operator experience did not affect the accuracy in some studies. Contrarily, in others, operator experience did affect the accuracy [18,19,22,23]. The operators had different levels of experience, and different study protocols were applied in those studies. Clinicians may have similar experience in scanning. However, it is not well known if the accuracy of their scans would still be similar. Therefore, it is essential to further investigate the various factors that affect the accurate transfer of the position of the implant, its relation with the remainder of the arch, and the scan accuracy of the entire arch, not just the implant’s itself.

Teeth act as fixed reference points and facilitate image acquisition and stitching during image acquisition [24]. Therefore, intraoral scans of a dentate arch have been reported to have acceptable accuracy [8,24]. However, conflicting results were reported for the accuracy of the scans when the scans were restricted to anterior or posterior regions of a dentate arch [25,26]. This could be attributed to the geometry difference in anterior and posterior teeth, which is known to affect the proper alignment [24,26], curvature of the arch [19,20,25,27] and the extent of the scanned area [23,25]. In a previous study, the accuracy tended to decrease from anterior to posterior teeth in the scans of a dentate arch [26]. Additionally, depending on the scanned region, complete- or partial-arch scans differed in accuracy of a dentate arch [28].

For single implant or prepared natural tooth scans, clinically acceptable accuracy was attributed to the limited extent of the scanned area when the arch was scanned partially [29,30,31]. Although complete-arch scans have the possibility of incurring misalignment errors with the increased scanned area and number of stitched images [17,20,22], complete-arch scans are still commonly performed for single implant scans [32]. Studies have shown that partial-arch scans can be as accurate as complete-arch scans, and recommended the use of partial-arch scans for implants both in the anterior and posterior regions [1,25,33,34]. However, those studies [1,25] focused on the accuracy of only the implant position. The scan accuracy of the remainder of the arch can also be crucial even though it may not directly affect the accuracy or the fit of an implant-supported crown. Clinicians primarily focus on the implant site during scanning, which may affect the accuracy of the rest of the arch. Dental arch accuracy may affect occlusion if deviations from the intraoral situation exist [25,33]. In addition, depending on the deviation’s magnitude and location, scan accuracy of the dental arch may affect the definitive restoration [33,35]. Therefore, the present study aimed to compare the trueness and precision of dental arches in scans during digitization of a single implant either with partial- or complete-arch scans. The accuracy of anterior and posterior sites was also aimed to be compared within and between the partial- and complete-arch scans when three different operators performed the scans. The null hypotheses were the following: (1) the scanned area (partial- vs. complete-arch) and the operator would not affect the accuracy of scans of the total scanned area; (2) the scan accuracy of the site (anterior vs. posterior left vs. posterior right in complete-arch and anterior vs. posterior left in partial-arch) would not be different for pooled data from all operators and for each operator, within each scan group; and (3) the accuracy of anterior sites and posterior sites would not be affected by the scan being complete- or partial-arch and the operator being also tested.

## 2. Materials and Methods

### 2.1. Data Acquisition

A partially edentulous maxillary model with an implant (4.0 × 11 mm) (Proactive Straight Implant; Neoss, Woodland Hills, CA, USA), and an intraoral scan body (Intra-Oral Scanbody, Neoss, Woodland Hills, CA, USA) at left central incisor site was additively manufactured (Form 2, Formlabs Inc., Somerville, MA, USA) [1]. A laboratory scanner with 4 µm accuracy was used (Ceramill Map 600, Amann Girrbach AG, Koblach, Austria) to obtain a reference scan of the model. Ten complete-arch scans (Figure 1) were performed with an IOS with 6.9 µm accuracy (TRIOS 3 v 1.4.7.5, 3Shape, Copenhagen, Denmark) by 3 operators with similar scanning experience (2 years of experience with intraoral scanning and at least 10 pilot scans with the IOS used). Partial-arch scans (Figure 2) were also performed by the same operators, from the distal of left 2nd molar to the distal of right canine (n = 10). The scan order was randomized by using a software program (Excel, Microsoft Corp, Redmond, WA, USA). A scan path, which was recommended by the manufacturer of the scanner, was followed for all scans, from the occlusal of left 2nd molars, continuing onto the occlusals/incisals of remaining teeth in each group followed by their lingual and buccal surfaces. All scans were performed in the same temperature-(20 °C) and humidity-controlled (45%) room, which was lit by sunlight and had an air pressure of 750 ± 5 mm [36].

### 2.2. Evaluation of Accuracy

The IOS’s scans were converted to standard tessellation (STL) files and exported to a 3-dimensional metrology software (Pro 8.1, GOM GmbH, Braunschweig, Germany) for comparisons with the reference scan. For the superimposition of reference and complete-arch scanned models, initially the GOM’s software prealignment feature was used, and all teeth and the scan body were selected for further alignment by using the “local best-fit” tool. On partial-arch scans, the area for superimpositions selected was from the distal of left 2nd molar to the distal of right lateral incisor. For evaluation, 4 planes in buccopalatal orientation were generated on the reference scan that passed through the distopalatal cusp of the 2nd molars and at the center of lateral incisors; and 3 planes in the mesiodistal orientation that passed through the center of the occlusal surface of 2nd molars, mesial of lateral incisors, and mesial of right central incisor. For complete-arch scans (Figure 3), 22 points were selected on 7 planes on 2nd molars and incisors at the gingival margin and the most incisal point both on buccopalatal and mesiodistal planes. For partial-arch scans (Figure 4), the mesiodistal plane and corresponding points on the right 2nd molar were not evaluated as that region was not captured with the IOS, resulting in 16 points on 6 planes. All coordinates for predefined planes and points were added, and the software algorithm generated the 3-dimensional (3D) variation between the points on the reference and the test scans.

### 2.3. Statistical Analysis

The data generated (3D distance deviation at all points, Figure 5) were used (Excel, Microsoft Corp, Seattle, WA, USA) for statistical analysis. The homogeneity of variances was analyzed by using Levene’s test, and a 2-way analysis of variance (ANOVA) followed by Tukey HSD were used to analyze the effect of the scanned area, the operator, and their interaction on trueness and precision of scans (α = 0.05).

## 3. Results

When the effect of the operator on the scanned area was considered, the two-way ANOVA revealed that only the scanned area affected the trueness of total area scanned (*p* < 0.001). Partial-arch scans had higher trueness than the complete-arch scans (*p* < 0.001). There was no significant difference among the operators in terms of the trueness of total scanned area in complete- (*p* ≥ 0.214) and partial-arch scans (*p* ≥ 0.073) (Table 1).

When the trueness was analyzed within scanned areas (complete- or partial-arch) (Table 2), for complete-arch scans, anterior sites had higher trueness than left posterior sites, which had higher trueness than right posterior sites overall (pooled data from all operators) (*p* < 0.001) and for each operators’ scans (for operator 1: *p* ≤ 0.022, for operator 2 and 3: *p* < 0.001) (Figure 6). For partial-arch scans, the anterior site had higher trueness than the left posterior site, overall (pooled data from all operators) (*p* < 0.001) and for all operators (*p* < 0.001 for operators 1 and 3, *p* = 0.016 for operator 2).

When the effect of the scanned area on trueness at anterior sites was considered by analyzing the pooled data from all operators (*p* = 0.33) and for each operator (*p* ≥ 0.056), no significant difference was found between partial- and complete-arch scans.

When the effect of the scanned area on trueness at left posterior sites was considered by analyzing the pooled data from all operators (*p* = 0.93) and for each operator (*p* ≥ 0.863), no significant difference was found between partial- and complete-arch scans.

For the precision of the total area scanned, the interaction between the operator and the scanned area was significant (*p* = 0.011). There was no significant difference among the operators in terms of total precision in complete-arch scans (*p* ≥ 0.294), whereas partial-arch scans of operator 1 had higher precision than the partial-arch scans of operator 2 (*p* = 0.036). No significant difference was observed in other operator pairs of partial-arch scans (*p* ≥ 0.207).

When the precision of complete-arch scans was considered by analyzing the pooled data from all operators, precision at anterior sites was higher than that at right and left posterior sites (*p* < 0.001), and no difference was found in precision for left and right posterior sites (*p* = 0.367). The same result was obtained when each operators’ scans were individually analyzed (*p* ≤ 0.003 for anterior vs. posteriors, *p* = 0.998 for operator 1, *p* = 0.494 for operator 2, and *p* = 0.252 for operator 3 for left to right comparison) (Figure 7).

When the precision of partial-arch scans was considered by analyzing the pooled data from all operators, the precision at anterior sites was higher than the precision at left posterior sites (*p* < 0.001). The precision was higher at the anterior site when the data were analyzed for each operator (*p* ≤ 0.001 for operator 1 and 3, and *p* = 0.016 for operator 2).

When the effect of the scanned area on the precision at anterior sites was considered by analyzing the pooled data from all operators (*p* = 0.124) and operator 3 (*p* = 0.334), no significant difference was found between partial- and complete-arch scans. For operator 1, the precision of anterior sites was higher in partial-arch scans than that at complete-arch scans (*p* = 0.008), whereas for operator 2, the precision of anterior sites was lower in partial-arch scans than that in complete-arch scans (*p* = 0.001).

When the effect of the scanned area on the precision at left posterior sites was considered by analyzing the pooled data from all operators, no significant difference was found between partial- and complete-arch scans (*p* = 0.974). The precision at left posterior sites was not significantly different between complete- and partial-arch scans when analyzed for each operator (*p* ≥ 0.906).

## 4. Discussion

The scanned area affected the trueness, and an interaction was found between the operator and the scanned area in terms of precision. Therefore, the first null hypothesis that the scanned area and the operator would not affect the accuracy of total scanned area was rejected. Scan accuracy in the anterior site was higher than that in the posterior site for both the complete- and partial-arch scans of all operators and their pooled data. Therefore, the 2nd null hypothesis was rejected. The operator and the scanned area affected the precision of the scans. Therefore, the 3rd null hypothesis that the accuracy of anterior and posterior sites would not be affected by the scan being complete-arch or partial-arch was rejected.

Direct comparison of the present study results with previous studies is difficult because previous studies focused on the accuracy of single implants (i.e., the scan bodies) [1,5] rather than the accuracy of scans of adjacent teeth and the arch. However, the positional accuracy of the arch is as important as that of the scan body for clinical success of implant-supported restorations. An inaccurately scanned arch may affect the 3D position of the restoration and its interproximal and occlusal contacts and contour [33,35]. Although the fit of the restoration may be acceptable, the restoration may need proximal and/or occlusal contact adjustments, including trimming or additional ceramic application, further glazing and/or polishing steps [33,35]. An over- or under-countoured restoration may affect the esthetics and hygiene, and thus require adjustments. When the restoration is made in monolithic form out of a CAD-CAM material and is under-contoured, additional ceramic or composite resin application on occlusal or proximal contacts may be problematic. Chipping can be a further clinical problem when ceramic or composite resin is applied over monolithic CAD-CAM ceramics or composite resins, since monolithic restorations are designed without cutback. The inaccuracy in the proximal contacts may affect the insertion axis of the crown, the path of withdrawal, and its proper seating. These additional adjustment steps may increase the time spent chairside and may decrease the efficacy of the direct digital workflow and comfort of the clinician and the patient. In addition, with screw-retained restorations, the location of the access hole within the overall crown shape may be improper, as the outer crown contours may deviate from the ideal. The above-mentioned issues may be encountered considering the high deviation values in the posterior region in the present study, and the deviation values exceed 1 mm in some areas.

Previously, the scans of incisal surfaces of the anterior teeth were reported to be difficult and more prone to errors during the alignment because of their simple geometries, whereas posterior teeth were reported easier to scan and align because of the complex geometries of occlusal surfaces on molars and premolars [24]. However, in the present study, higher accuracy was obtained in the anterior site than in the posterior in complete- and partial-arch scans. The presence of the scan body, which has a wider upper surface area compared to the incisal surfaces of the incisors, might have helped for the proper alignment of scans in the anterior site and decreased the deviation. Another contributing factor for increased accuracy in the anterior region may be the linear scanning path of the arch in the anterior region [27]. The scans of the implant used in the present study, which was located in the anterior region, has been reported to be acceptable.

In complete-arch scans, left posterior sites had higher trueness than the right posterior sites, whereas their precision was similar. In the present study, the scans started from the left side of the model. Therefore, lower trueness with the right side may be attributed to the increased number of stitched images and the possibility of increased inherent errors and misalignment of errors when the scanned area increased [23]. Low accuracy was reported in previous studies in the extended scanned areas [23,25,28].

In the present study, trueness at anterior and left posterior sites and the precision at left posterior sites was similar between complete- and partial-arch scans independent of operator. However, while the partial-arch scans of operator 1 had higher precision than that of complete-arch scans at the anterior site, the exact opposite was observed for operator 2. Given that operator 1 also had higher scan precision than operator 2 when partial-arch scans were considered, this result seems consistent. Even though the operators in the present study had similar experience, more reliable results can be obtained and any significant differences among operators may be elaborated thoroughly with the involvement of more operators. Nevertheless, many previous studies used three operators to investigate the effect of the operator on scan accuracy [21,23,37].

The present study was performed under standardized laboratory conditions without including patient specific factors, and the intraoral conditions and results may differ if patient-specific factors are involved [17,35]. In addition, considering that scan accuracy of IOSs is also affected by environmental conditions [1], other in vitro studies might have different results. Previous studies reported that IOS technology and scan path might affect the accuracy of implant scans [17,20], thus the findings may vary depending on the scanner technology and scan path. The use of a laboratory scanner to obtain the reference scans is a limitation. However, the accuracy of the applied laboratory scanner is high, and the application of laboratory scanners to obtain reference datasets has been recommended [38,39,40,41]. Future studies should be performed by using an industrial grade scanner for high-accuracy reference scans. The present study results should be validated further by fabricating definitive single unit implant crowns and evaluating their relationship with adjacent teeth and antagonists. Future clinical studies are necessary to corroborate the findings of the present in vitro study.

## 5. Conclusions

The accuracy of intraoral scans obtained with the IOS used was significantly affected by the scans being partial- or complete-arch. Partial-arch scans had higher trueness than the complete-arch scans, but their precision was similar. The accuracy of the total scanned area was similar among different operators’ scans. When the accuracy of scans at different sites in the arch is considered, anterior sites had higher accuracy compared to posterior sites for both partial- and complete-arch scans. In terms of the operator’s effect on trueness and precision of scans at different sites in the arch, significant differences were only observed in the precision of anterior site scans of two operators. The scan accuracy of the operators was similar for the remainder of the sites evaluated.

## Figures and Tables

**Figure 1 jcm-11-04125-f001:**
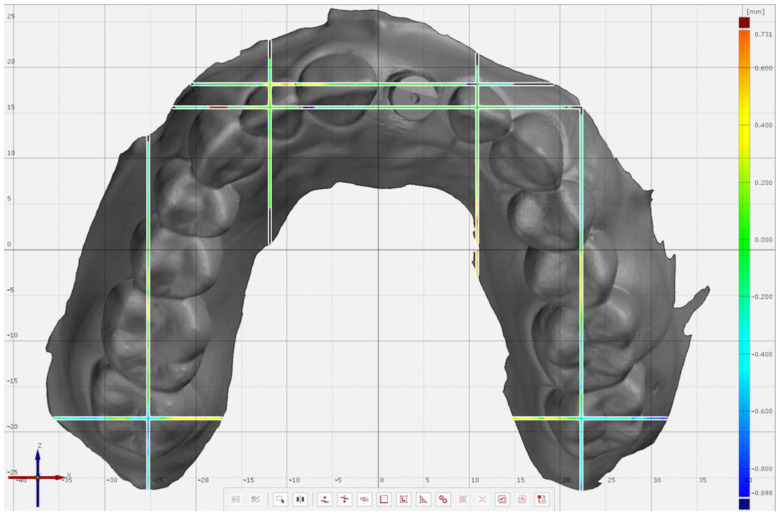
Complete-arch intraoral scan including the planes for subsequent analyses.

**Figure 2 jcm-11-04125-f002:**
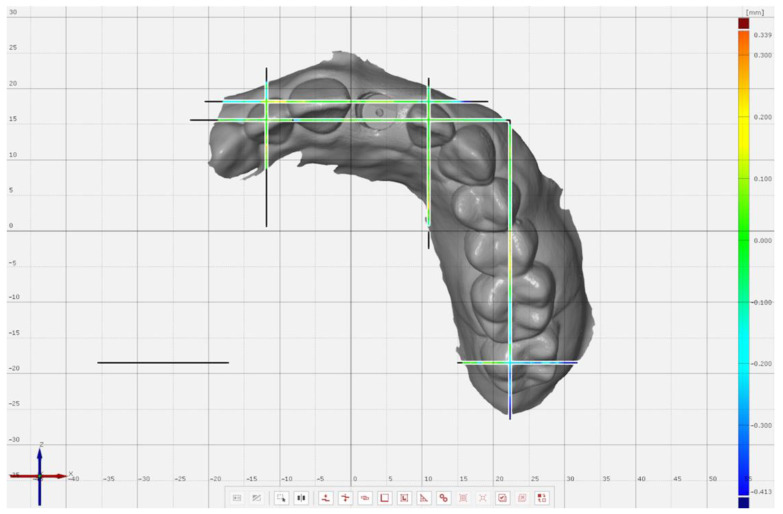
Partial-arch intraoral scan, including the planes for subsequent analyses.

**Figure 3 jcm-11-04125-f003:**
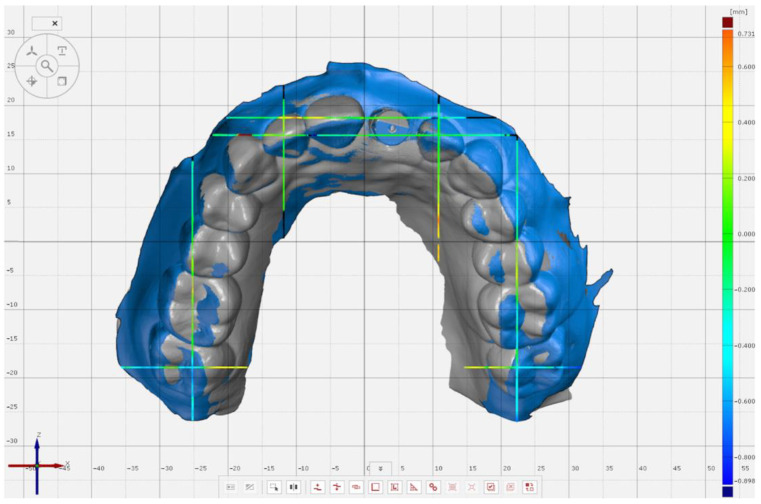
Measurement points (Left 2nd molar: Buccal, palatal, and distal gingival margins, mesiopalatal and distopalatal cusp tips, and most buccal point of the tooth. Left lateral incisor: Buccal, palatal, and mesial gingival margins, most mesial and incisal point of the tooth. Right central incisor: Mesial gingival margin and most mesial point of the tooth. Right lateral incisor: Buccal and gingival margins and most incisal point of the tooth. Right 2nd molar: Buccal, palatal, and distal gingival margins, mesiopalatal and distopalatal cusp tips, and most buccal point of the tooth) on complete-arch scan at selected planes when superimposed to the reference scan.

**Figure 4 jcm-11-04125-f004:**
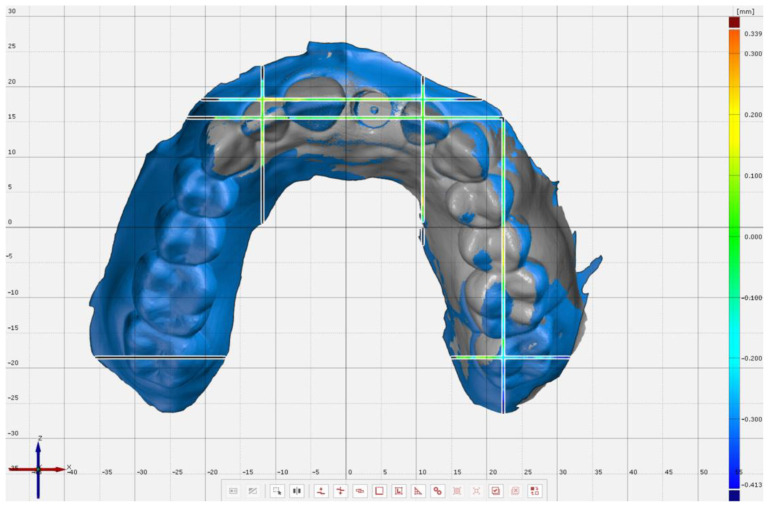
Measurement points (Left 2nd molar: Buccal, palatal, and distal gingival margins, mesiopalatal and distopalatal cusp tips, and most buccal point of the tooth. Left lateral incisor: Buccal, palatal, and mesial gingival margins, most mesial and incisal point of the tooth. Right central incisor: Mesial gingival margin and most mesial point of the tooth. Right lateral incisor: Buccal and gingival margins and most incisal point of the tooth) on partial-arch scan at selected planes when superimposed to the reference scan.

**Figure 5 jcm-11-04125-f005:**
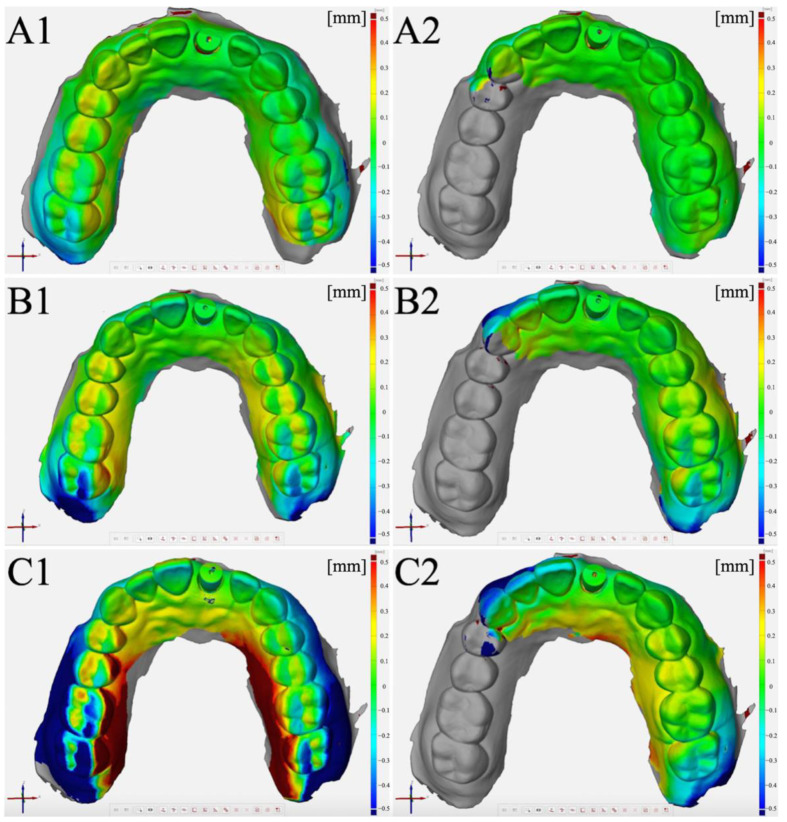
Color maps generated by the superimposition of test scans over reference scan: (**A**) Operator 1. (**B**) Operator 2. (**C**) Operator 3. (**1**) Complete-arch scan. (**2**) Partial-arch scan.

**Figure 6 jcm-11-04125-f006:**
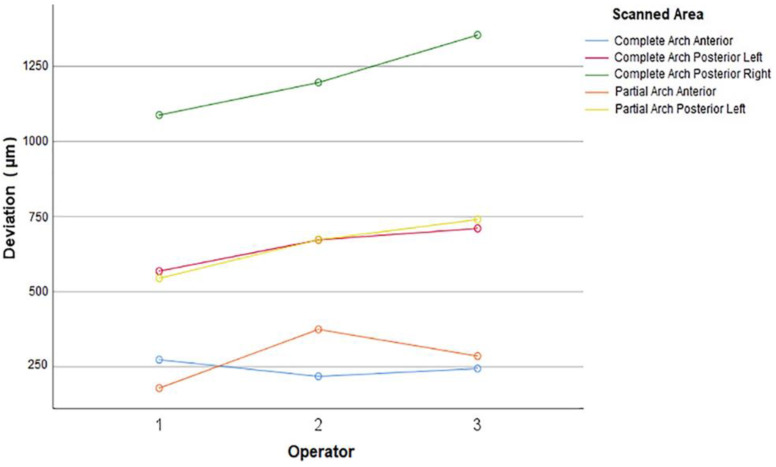
Deviations (indicating trueness) of different sites in complete- and partial-arch scans for each operator.

**Figure 7 jcm-11-04125-f007:**
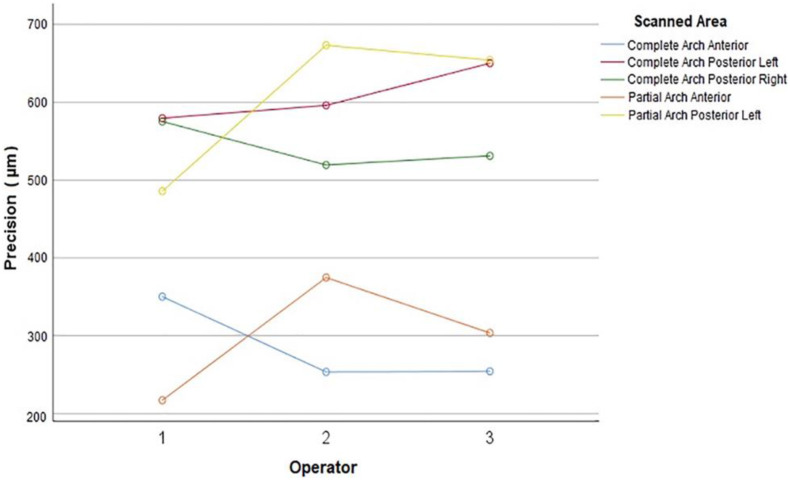
Precision of scans at different sites in complete- and partial-arch scans for each operator.

**Table 1 jcm-11-04125-t001:** Two-way ANOVA results for trueness and precision.

Property	Effect	Df	F-Ratio	*p*-Value
	Operator	2	2.671	0.070
Trueness (µm)	Scanned area	1	24.706	<0.001
	Operator × Scanned area	2	0.763	0.467
	Operator	2	1.067	0.344
Precision (µm)	Scanned area	1	1.379	0.241
	Operator × Scanned area	2	4.505	0.011

df, numerator degrees of freedom.

**Table 2 jcm-11-04125-t002:** Mean and standard deviation values for trueness and precision of complete and partial arch scans from different operators and pooled data from all operators.

Parameter	Arch	Area	Operator 1	Operator 2	Operator 3	Operator Pooled
Trueness (µm)	Complete	Ant	273.2 ± 544.9 ^A,1^	217.9 ± 404.3 ^A,1^	243.9 ± 420.8 ^A,1^	245.3 ± 462.0 ^A,1^
		Left Post	568.2 ± 735.2 ^B,2^	672.3 ± 767.9 ^B,2^	710.3 ± 831.6 ^B,2^	648.2 ± 774.3 ^B,2^
		Right Post	1088.4 ± 667.4 ^C^	1196.4 ± 596.6 ^C^	1354.7 ± 604.9 ^C^	1208.3 ± 630.2 ^C^
		Total	583.7 ± 717.87 *	621.0 ± 704.4 *	686.3 ± 760.0 *	628.3 ± 726.6
	Partial	Ant	178.9 ± 287.5 ^D,1^	374.2 ± 676.1 ^D,1^	285.4 ± 457.7 ^D,1^	286.1 ± 515.6 ^D,1^
		Left Post	544.5 ± 740.8 ^E,2^	673.0 ± 766.7 ^E,2^	739.8 ± 827.2 ^E,2^	656.1 ± 779.1 ^E,2^
		Total	310.5 ± 527.6 *	475.2 ± 719.5 *	443.2 ± 646.4 *	414.8 ± 643.7
Precision(µm)	Complete	Ant	350.1 ± 415.7 ^A,1^	253.4 ± 311.7 ^A,1^	254.2 ± 332.2 ^A,1^	287.3 ± 358.9 ^A,1^
		Left Post	579.6 ± 444.3 ^B,3^	596.1 ± 476.2 ^B,3^	650.2 ± 509.0 ^B,2^	607.2 ± 473.7 ^B,2^
		Right Post	575.4 ± 330.6 ^B^	519.5 ± 286.0 ^B^	531.3 ± 279.6 ^B^	542.4 ± 299.5 ^B^
		Total	472.0 ± 414.2 *	416.5 ± 382.8 *	432.4 ± 406.0 *	440.6 ± 401.1
	Partial	Ant	217.1 ± 186.7 ^C,2^	374.7 ± 676.2 ^C,2^	303.5 ± 340.8 ^C,1^	334.8 ± 383.8 ^C,1^
		Left Post	485.8 ± 666.6 ^D,3^	673.2 ± 766.9 ^D,3^	654.1 ± 496.8 ^D,2^	612.6 ± 478.6 ^D,2^
		Total	313.8 ± 443.4 *	475.5 ± 719.6 ^ǂ^	425.2 ± 433.8 *^,^^ǂ^	409.7 ± 557.3

Ant, anterior; post, posterior. Significant differences among scanned sites are presented by using different uppercase superscript letters in the same column for complete- and partial-arch, independently. Significant differences between the same scanned sites (anterior or left posterior) of partial- and complete-arch scans are presented by using different numbers in the same column. Significant differences between total trueness and precision of operator groups of complete- and partial-arch scans are presented by using different symbols in the same row.

## Data Availability

Data sharing is not applicable for this paper.

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
