# Peer review of "Effect of Scanned Area and Operator on the Accuracy of Dentate Arch Scans with a Single Implant"

_jcm, 2022, doi:10.3390/jcm11144125_

Round 1

Reviewer 1 Report

  • This laboratory study is investigating the effect of the arch on the accuracy, precision, and trueness of single implant scans via a known intra-oral scanner. The methods are straightforward and thoroughly presented. My only concern is that the test subject and the scanning environment are not clearly described. Since it is known that the OI scanners interact with various environmental variables these should be demonstrated in the material and methods by additional photographs. If the test subject/environment is not comparable to previous studies the implications should be additionally discussed.

Author Response

This laboratory study is investigating the effect of the arch on the accuracy, precision, and trueness of single implant scans via a known intra-oral scanner. The methods are straightforward and thoroughly presented. My only concern is that the test subject and the scanning environment are not clearly described. Since it is known that the OI scanners interact with various environmental variables these should be demonstrated in the material and methods by additional photographs. If the test subject/environment is not comparable to previous studies the implications should be additionally discussed.

Response: The authors would like to thank Reviewer #1 for their comments. Test subject and the scanning environment are now elaborated in Material and Methods section. The scans were performed in a humidity and temperature-controlled room. The model used and scanning conditions in the present study were similar to those of a previous study on the scan accuracy of a single anterior implant (Reference #1), therefore, comparisons are possible and comparisons with previous studies have been made. The authors are not certain whether including the picture of the laboratory the scans were performed would be of considerable addition to the scientific quality of the paper. We are happy to add one if the editor would encourage so. Nevertheless, the fact that these environmental conditions might affect the scan accuracy is now mentioned in the final paragraph of Discussion.

Reviewer 2 Report

The article investigate on an interesting topic as the Intraoral scanner are nowadays commonly used. 

The english language is fine

The introduction provide sufficient number and quality of references,

The methodology is correctly described.

It is suggested to indicate the software version of the IOS used

In the discussion A.A. could define what could be defined as clinically acceptable.

In the discussion the A.A. should further discuss their findings .

Moreover it could be interesting for the readers to have some clinically applicable information as Giudice RL, Famà F. Health care and health service digital revolution. Int J Environ Res Public Health 2020;17(14):1-2.

Author Response

The article investigate on an interesting topic as the Intraoral scanner are nowadays commonly used. The English language is fine. The introduction provides sufficient number and quality of references. The methodology is correctly described.

Response: The authors would like to thank Reviewer #2 for their comments and contributions to the scientific quality of our paper.

It is suggested to indicate the software version of the IOS used.

Response: The software of the IOS used in the present study is now mentioned in Material and Methods.

In the discussion A.A. could define what could be defined as clinically acceptable. In the discussion, the A.A. should further discuss their findings.

Response: The authors would like to thank the reviewer for their comment. The definition of a clinically acceptable deviation only considering the results of the present study is difficult. Even though the accuracy of the scan of the remaining arch would affect the clinical outcomes (interproximal and occlusal contacts) with a definitive restoration, these outcomes may also be affected by other factors such as manufacturing quality or the restorative material itself. These details have been emphasized in the second paragraph of Discussion. Nevertheless, it is also mentioned in the manuscript that restoration would potentially need further adjustments when high deviation values in the posterior region in the present study are considered, particularly for those situations where deviation values exceed 1 mm in some areas. The authors believe that the findings of the present study are elaborated in terms of their potential clinical effects in Discussion, which is almost 3-pages long.

Moreover, it could be interesting for the readers to have some clinically applicable information as Giudice RL, Famà F. Health care and health service digital revolution. Int J Environ Res Public Health 2020;17(14):1-2.

Response: Mentioned paper is cited in Introduction (Reference #6) and the first sentence of Introduction is elaborated accordingly. The sentence now reads “The launch of new intraoral scanners (IOSs) and improved accuracy with scanner technologies enabled the fabrication of implant-supported monolithic crowns through a direct digital workflow with clinically acceptable accuracy and less patient discomfort in recent years [1-7].”

This manuscript is a resubmission of an earlier submission. The following is a list of the peer review reports and author responses from that submission.

Round 1

Reviewer 1 Report

Thank you for your work. Good topic, and generally I’m generally happy to read the content of the manuscript but some minor points need clarifying before publish.

Abstract

please double check IJERPH style: trueness (P≤.022, precision: P≤.003, ….) Pls add the exact P-value

Intro

- Objectives would seem well-elaborated.

- Pls describe about trueness of IOS

Mat & Methods

- Pls clearly describe the planes for reference scan. In addition, please clearly state the analyzed points used in Figure 3 and Figure 4.

- Pls clearly identify how to analyze trueness and precision?

Results

- Where appropriate, please add exact p values with your text, please double check IJERPH style

- Do you have any strong reason for the differences between the operators?

Discussion

Pls, discuss about the dental arch accuracy affecting the occlusion as the current results revealed that the deviation is prominent in the posterior region although this study was performed under lab conditions?

Pls double-checked at line no 242?

Reviewer 2 Report

The number of operators (3) is not sufficient to define a complete statistical analysis.

The operator's skills and experience are not sufficiently and quantitatively defined to perform a comparative analysis.

The statistical analysis could have more significance if referred to well-defined levels (to be clearly defined) of skill of the operators to be grouped for the subsequent different analyses.

Reviewer 3 Report

The manuscript compares the trueness and precision of dental arches in scans during digitization of a single implant either with partial- or complete-arch scans. Considering the recent booming of digital dentistry, the manuscript may be of great interest to the relevant communities. However, the manuscript has serious flaws in research design and interpretation, particularly because of their small sample size – the study involved only 3 operators for performing statistical analysis and drawing conclusions. The authors should justify or reconsider if such small sample size is acceptable. (If justifiable, elaborate in the manuscript.) Also, the legends in Figure 5 are too small to read.